# Meconium Aspiration Syndrome: A Narrative Review

**DOI:** 10.3390/children8030230

**Published:** 2021-03-17

**Authors:** Chiara Monfredini, Francesco Cavallin, Paolo Ernesto Villani, Giuseppe Paterlini, Benedetta Allais, Daniele Trevisanuto

**Affiliations:** 1Neonatal Intensive Care Unit, Department of Mother and Child Health, Fondazione Poliambulanza, 25124 Brescia, Italy; chiara.monfredini@poliambulanza.it (C.M.); paolo.villani@poliambulanza.it (P.E.V.); giuseppe.paterlini@poliambulanza.it (G.P.); benedetta.allais@poliambulanza.it (B.A.); 2Independent Statistician, 36020 Solagna, Italy; cescocava@libero.it; 3Department of Woman and Child Health, University of Padova, 35128 Padova, Italy

**Keywords:** infant newborn, meconium aspiration syndrome, meconium-stained amniotic fluid

## Abstract

Meconium aspiration syndrome is a clinical condition characterized by respiratory failure occurring in neonates born through meconium-stained amniotic fluid. Worldwide, the incidence has declined in developed countries thanks to improved obstetric practices and perinatal care while challenges persist in developing countries. Despite the improved survival rate over the last decades, long-term morbidity among survivors remains a major concern. Since the 1960s, relevant changes have occurred in the perinatal and postnatal management of such patients but the most appropriate approach is still a matter of debate. This review offers an updated overview of the epidemiology, etiopathogenesis, diagnosis, management and prognosis of infants with meconium aspiration syndrome.

## 1. Definition of Meconium Aspiration Syndrome

Meconium aspiration syndrome (MAS) is a clinical condition characterized by respiratory failure occurring in neonates born through meconium-stained amniotic fluid whose symptoms cannot be otherwise explained and with typical radiological characteristics [1]. The severity of MAS can be defined as mild (FiO2 < 0.40 for less than 48 h), moderate (FiO2 > 0.40 for more than 48 h without air leak) or severe (mechanical ventilation for more than 48 h and/or pulmonary hypertension) according to Clearly and Wiswell [2].

## 2. Epidemiology

As meconium is uncommon in amniotic fluid before 34 weeks’ gestation, MAS is a typical disease of near-term, term or post-term newborns. Meconium-stained amniotic fluid (MSAF) is found in 4–22% of all births [3]; up to 23–52% in those beyond 42 weeks’ gestation [4]. Only 3–12% of the babies born through MSAF develop MAS. Among them, 20% are non-vigorous at birth, about one third requires intubation and mechanical ventilation [4] and 5–12% die [5]. Overall, MAS explains about 10% of neonatal respiratory failure [6]. The incidence of MAS exponentially increases from 38 to 42 weeks’ gestation [3,7]. Worldwide, the incidence of MAS has declined in developed countries thanks to improved obstetric practices and perinatal care while challenges persist in developing countries [8].

## 3. Etiopathogenesis

Meconium is a black-greenish, sterile, odorless, dense, sticky and viscous material consisting of water, skin and intestinal desquamation cells, gastrointestinal secretions, bile, pancreatic juices, mucus, lanugo, vernix, blood glycoproteins and amniotic fluid [9]. The term meconium is derived from the Greek “mekoni” (poppy juice or opium) because of its black appearance [9].

Meconium is usually found in the intestine of the fetus and newborn with its first emission occurring within the first day of life. However, meconium can sometimes be released early into the amniotic fluid causing the so-called “stained fluid” [9]. While meconium appears in the intestine at around 12 weeks’ gestation, the passage into the amniotic fluid is unlikely in preterm infants because of ineffective peristalsis, good anal sphincter tone and low motilin levels. When found in a preterm infant, biliary discoloration (due to intestinal obstruction) or fetal diarrhea (secondary to sepsis, particularly from *Listeria*) should also be considered [10].

The presence of MSAF is considered to be an indicator of fetal stress due to hypoxia and acidosis causing a vagal response that triggers an increase in peristalsis and anal sphincter release with a resulting meconium passage into the uterine cavity [10,11]. Risk factors for MSAF include placental insufficiency, maternal hypertension, pre-eclampsia, oligohydramnios with umbilical cord compression during labor, infections and maternal substance abuse (especially nicotine and cocaine) [11].

MAS results from the aspiration of MSAF during gasping in intrauterine life or during the first breaths after birth [1,2]. Risk factors for MAS include thick meconium, a pathological cardiotocographic trace, fetal acidosis, caesarean section, the need for intubation at birth and low Apgar scores [1,2].

## 4. Pathophysiology

MAS has a multifactorial pathophysiology (Figure 1). The main pathophysiological mechanisms include:(a)Antenatal inflammation/infection [12,13,14,15]: as bacteria, endotoxin and high concentrations of inflammatory mediators have been found in MSAF, a fetus swallowing such microbial products and inflammatory mediators can experience increased intestinal peristalsis and passage of meconium, which can be aspirated by the fetus. A recent study reported the presence of meconium in the alveoli of stillbirths, which suggested an antemortem meconium passage in utero due to hypoxia and inflammatory processes. Further, histological findings showed an increased acute placental inflammation in MSAF. Although amniotic fluid has bacteriostatic properties, the addition of a small amount of meconium impairs its inhibitory effect and can enhance the growth of bacteria such as group B streptococcus and *Escherichia coli*.(b)Mechanical airway obstruction [4,7]: the occlusion of the airways by meconium plugs leads to a high resistance to air flow and air trapping according to the consistency and quantity of the meconium-stained liquid. If the obstruction is partial, valve effects lead to a hyperinflation condition; if the obstruction is total, “patchy” areas of atelectasis are caused. Trapped gas can lead to air leak such as interstitial emphysema, pneumothorax and pneumomediastinum. Partial or complete airway obstructions have been considered to be the main pathophysiological mechanism of MAS for many years.(c)Inactivation of the surfactant [4,7]: the surfactant inactivation due to the action of meconium fatty acids causes atelectasis and impairs the ventilation-perfusion mismatch. Although the precise mechanism is not fully understood, fat-soluble and water-soluble components of the meconium seem to be involved in this process. Meconium is able to alter the viscosity and ultrastructure of the surfactant through direct toxicity on type II pneumocytes. Furthermore, it reduces the levels of proteins A and B and accelerates the conversion of active large aggregates into less active smaller forms and determines the displacement from the alveolar surface. Surfactant dysfunction is further worsened by binding to plasma proteins due to damage of the alveolar-capillary membrane and the presence of proteolytic enzymes and oxygen free radicals.(d)Activation of the inflammatory cascade [4,16,17]: the alveolar interstitium of patients with MAS shows inflammatory cellular infiltrates characterized by the release of cytokines and complement activation. Meconium contains substances with a chemotactic action for neutrophils; it also activates the complement, has a vasoactive function and is also a source of pro-inflammatory mediators (such as IL-1, IL-6 and IL-8 and TNF). Despite the repairing role of inflammation, its destructive potential can cause local tissue damage. For decades it has been widely known that meconium is toxic and induces inflammation and apoptosis and can lead to chemical pneumonia in the first 48 h of life with a risk of bacterial over-infection. However, the cellular mechanism underlying the initiation of the inflammatory cascade in humans remains to be clarified. As meconium is produced in the intestine and is therefore only minimally exposed to the immune system during fetal life, it may be recognized as “not self”, triggering the activation of innate immunity. It has been hypothesized that the two main systems of the recognition of innate immunity (the toll-like receptor and the complement system) may recognize meconium as dangerous and activate the inflammatory cascade. In vivo, it is reasonable to hypothesize that additional triggers for inflammation can be hypoxia due to MAS, baro- and volu-trauma related to ventilation and oxygen therapy. Understanding the mechanisms underlying the inflammatory cascade in MAS could be useful for addressing new therapeutic strategies.(e)Persistent pulmonary hypertension [4]: it occurs in 15–20% of MAS patients and has been linked to different mechanisms including pulmonary vasoconstriction (secondary to hypoxia/hypercapnia/acidosis), capillary hypertrophy (due to intrauterine hypoxia) and pulmonary hyperexpansion (increasing pulmonary resistance). The right-left shunts worsen the hypoxemia and can lead to a dangerous vicious circle.

## 5. Factors Associated with MAS

Several studies have investigated antenatal and perinatal factors associated with MAS to improve the identification and management of such patients. Fetal distress or non-reassuring/abnormal cardiotocography have been frequently reported in MAS patients (43–67%) [18,19,20,21,22,23] as well as umbilical cord metabolic acidosis (21–24%) [18,19,20] and a low Apgar score at 5 min (18–60%) [18,19,20,21,22,23]. In addition, Karabayir et al. suggested that increased lactate levels in blood gases during the first hour of life may also be a risk factor for the development of MAS in MSAF patients [24]. The presence of inflammation/infection in maternal history has been associated with MAS. Funisitis and/or acute chorioamnionitis have been reported in 18–65% of MAS patients [18,19,20,22,23], intrapartum fever in 49–57% [18,19] and a rupture of the membrane > 18 h in 5–30% [22,23]. Of note, Gupta et al. showed an increased risk of MSAF and MAS among infants born to HIV-positive mothers although the effects of maternal HIV infection and anti-retroviral therapy remain unclear [25]. Conflicting results have been reported about the association between caesarean section and MAS and thus it remains controversial whether birth by caesarean section may be accounted among the risk factors for MAS [19,20,22,23,26,27,28,29]. The consistence of meconium seems to be another important indicator of MAS as thick meconium has been reported in 51–75% of MAS patients [23,30]. Despite recent decreasing rates due to changes in official recommendations on neonatal resuscitation (Section 11), a large proportion of MAS patients (35–65%) still needs endotracheal intubation at birth [20,22,31]. Of note, available evidence suggests that the development of severe MAS should not be considered as a “continuum” of the same risk factors causing mild/moderate MAS [19].

## 6. Clinical Features

The first clinical sign of MAS is the presence of MSAF at birth in a non-vigorous infant, suggesting the typical pattern of asphyxia [32]. The general findings may also include (hypoxic-ischemic) encephalopathy, heart failure, poor peripheral perfusion and a reduction of urine output. A neonate with MAS shows respiratory distress with heterogeneous severity associated with tachypnea, cyanosis, nasal flaring, respiratory retractions and a hyperexpanded and barrel-shaped thorax. Widespread crackles are found on auscultation. Newborns with MAS can enter a dangerous vicious circle: hypoxemia leads to acidosis and both determine a worsening of pulmonary hypertension; the pulmonary hypertension causes a right-to-left shunt at the level of the foramen ovale and the ductus arteriosus thus causing cyanosis and hypoxemia that continue the vicious circle [33]. Respiratory findings range from mild/moderate respiratory distress to severe refractory hypoxemia secondary to persistent pulmonary hypertension (PPHN), which requires advanced respiratory support (such as high frequency oscillatory ventilation, inhaled nitric oxide and extracorporeal membrane oxygenation) [34].

## 7. Diagnosis

The following criteria for the diagnosis of MAS have been suggested [16,35]:Respiratory distress in a newborn born through MSAF;Oxygen requirement to maintain transcutaneous saturation over 92%;The need for oxygen therapy within 2 h of life and for at least 12 hThe absence of malformations of the airways, lungs and heart.

The diagnosis of MAS is based on maternal history (full-term or post-term pregnancy, perinatal distress, the presence of MSAF), clinical features (a full-term or post-term newborn with meconium painted skin and respiratory distress characterized by the hyperexpansion of the thorax) and a chest X-ray (pulmonary hyperinflation with cottony and patchy infiltrates alternating with areas of hypertransparency; Figure 2).

A lung ultrasound has been increasingly used as a diagnostic and prognostic technique in the neonatal intensive care unit [36,37]. A few dynamic lung ultrasound signs (B-pattern interstitial coalescent or sparse consolidations, atelectasis, bronchograms) have been observed in MAS patients thus suggesting its possible use in clinical practice [38]. While potentially reducing the use of X-rays, a lung ultrasound should not completely replace a standard chest X-ray and should be considered alongside the perinatal history [38].

An echo-doppler should be routinely used to investigate the presence of PPHN in MAS patients. The main hemodynamic features of PPHN include: (i) a decreased pulmonary flow with a ventilation-perfusion mismatch; (ii) systo-diastolic right ventricular (RV) dysfunction (caused by an increased afterload); (iii) decreased RV stroke volume and decreased RV filling; (iv) RV dilatation with a D-shape; (v) decreased left ventricular (LV) stroke volume with systemic hypotension; (vi) systo-diastolic LV dysfunction (caused by a decreased preload); (vii) right-to-left shunting through the foramen ovale and Botallo ductus [39,40].

Of note, the severity of the radiological picture does not always correlate with the clinical severity [4] thus suggesting that the severity of MAS is due to other factors (such as pulmonary hypertension) beyond the degree of airway obstruction and parenchymal damage.

Respiratory severity may be assessed by using the oxygenation index (OI) calculated as OI = (mean airway pressure × FiO_2_ × 100)/preductal PaO_2_ [34]. The severity ranges from mild (OI < 15) to moderate (OI 15–25) to severe (OI 25–40) and up to very severe (>40) [41]. An OI over 40 for more than 4 h is included among the indications for initiating extracorporeal membrane oxygenation [42]. When a preductal gas analysis is not available, the OI can be estimated by using the oxygen saturation index (OSI) (OI ≈ 2 × OSI) calculated as OSI = (mean airway pressure × FiO_2_ × 100)/preductal SpO_2_ [43].

As MAS is frequently associated with sepsis, blood work (including a white blood count and differential, a platelet count, C-reactive proteins, procalcitonin) and culture of blood, spinal fluid, urine, gastric aspirate and tracheal aspirate should be obtained [44,45].

## 8. Treatment

The goals of MAS treatment include general aspects (preventing or resolving infection and metabolic disorders, preventing hypoxic-ischemic brain injury and reducing stress), respiratory aspects (optimizing lung ventilation, promoting pulmonary vasodilation and enhancing oxygenation while preventing iatrogenic damage) and hemodynamic stabilization.

### 8.1. General Treatment

All infants born through MSAF showing respiratory distress must be admitted to the neonatal intensive care unit (NICU) where they can be closely monitored. Maintaining normothermia (36.5 and 37.5 °C) should be warranted in all infants apart from asphyxiated infants requiring therapeutic hypothermia. A parenteral nutrition solution (with glucose, amino acids, lipids and electrolytes) should be provided.

Metabolic acidosis and any metabolic disorders (such as hypoglycemia) should be promptly corrected [4,16]. The goal is to maintain pH in the range 7.25–7.40 and PaCO_2_ in the range 40–55 mmHg. Caregivers should avoid hyperventilation causing respiratory alkalosis and sodium bicarbonate infusion causing metabolic alkalosis, which can increase the risk of neurodevelopmental impairment [41].

Noise, tactile and light stimuli should be minimized to prevent pulmonary hypertension. Intubated MAS patients require sedation and analgesia to reduce discomfort and, therefore, the right-to-left shunt that further aggravates hypoxemia [4,10]. Opioids (such as fentanyl and morphine) are the most common sedation drugs, reducing asynchrony and discomfort and, as a result, improving gas exchange. Neuromuscular blockers (such as pancuronium and vecuronium) were broadly used in combination with opioids in the past but their use is controversial and should be reserved for cases with an insufficient response to sedation. In fact, neuromuscular blockers favor the pulmonary atelectasis, cause ventilation-perfusion mismatch and are associated with a higher mortality risk [10].

Meconium is sterile but prone to over-infection especially in the areas of the lung that are not adequately ventilated. Antibiotic prophylaxis should not be recommended in MAS patients because there is no evidence supporting the hypothesis of the association between MAS and sepsis [45,46]. However, MAS is clinically diagnosed and radiological findings cannot exclude pneumonia [7,11]. A broad-spectrum antibiotic therapy should be initiated in the presence of perinatal infectious risk factors and suspended after 48–72 h if the blood culture is negative [47].

### 8.2. Respiratory Support

Respiratory support ranges from oxygen therapy (for the mildest forms) to non-invasive ventilation (for the moderate forms) up to mechanical ventilation (for the most severe cases).

Oxygen therapy should aim to maintain preductal saturation between 92–97% (PaO_2_ between 50–80 mmHg) because hyperoxia can exacerbate arterial pulmonary vasoconstriction and impair the response to inhaled nitric oxide (iNO) [41].

Nasal continuous positive airway pressure (NCPAP) has proven superior in avoiding mechanical ventilation compared with oxygen therapy alone [48]. NCPAP should be set at 6–8 cm H_2_O and aimed at optimizing lung recruitment as shown by the expansion of approximately 8–9 ribs on an anteroposterior chest X-ray.

About 40% of infants with MAS require mechanical ventilation [49]. In our institution, intubation is performed based on pH and PaCO_2_ values (pH < 7.25 and PaCO_2_ > 60 mmHg). The ventilatory management of these newborns is particularly complex because of the alternation of atelectatic (difficult to recruit) and hyperinflated (at risk of air leak) areas [4]. Therefore, the pressure values must be set individually; these infants often require high inspiratory pressures (although it is always desirable not to exceed 25 cm H_2_O) and large tidal volumes [48] with positive end expiratory pressure (PEEP) that should be maintained between 4–6 cm H_2_O to avoid alveolar hyperdistention. The expiratory times must be long enough to avoid air trapping patterns (to which these infants are susceptible), which can result in ineffective ventilation and air leaks (i.e., pneumothorax, pneumomediastinum).

High frequency oscillatory ventilation (HFOV) is indicated to reduce the barotrauma, guarantee a more homogeneous recruitment and prevent the risk of air leaks [34]. Switching to HFOV can be considered when the peak inspiratory pressure is > 25–28 cm H_2_O. Unfortunately, there are no randomized controlled trials (RCT) assessing different ventilation strategies in MAS patients.

### 8.3. Surfactant

The Committee on Fetus and Newborn of the American Academy of Pediatrics recommends surfactant administration in MAS patients because it improves oxygenation and reduces the need for extracorporeal membrane oxygenation (ECMO) [50]. The Canadian Pediatric Society recommends surfactant therapy for all intubated infants with MAS and oxygen requirements greater than 50% [51]. The surfactant can be administered as a bolus or bronchial lavage.

Bolus surfactant therapy improves the respiratory function and oxygenation index within 6 h after treatment in infants with MAS. Although reducing the need for ECMO and MAS severity, there is no evidence of the benefits regarding mortality, air leak, duration of ventilation, incidence of chronic lung disease or intraventricular hemorrhage [52]. The effectiveness of bolus surfactant therapy compared with or in combination with other therapies (such as inhaled nitric oxide, lung lavage, HFOV) merits further investigation in infants with MAS.

Therapeutic lung lavage can be described as “any procedure in which fluid is instilled into the lung, followed by an attempt to remove it by suctioning and/or postural drainage” [53]. In human infants, lung lavage dates to the early 1970s but was later abandoned due to the increase of newborns with transient tachypnea imputed to the retention of the washing fluid [54,55]. Improvements in oxygenation and carbon dioxide clearance after lung lavage were then described in isolated reports in the 1990s [56,57]. More recent investigations suggested that surfactant lung lavage may be beneficial for infants with MAS but more research on the method, comparisons with other approaches and long-term outcomes are warranted [58].

### 8.4. Inhaled Nitric Oxide

MAS patients with persistent pulmonary hypertension should receive iNO therapy while ensuring adequate sedation and maintaining sub-systemic pulmonary pressures [59]. iNO therapy reduces need for ECMO and mortality in full-term or near-term infants with respiratory failure and persistent pulmonary hypertension [60] and its effect is enhanced when using HFOV as a ventilatory strategy [61]. In a MAS patient with an oxygenation index reaching 15–25, iNO therapy should be started at an initial dose of 20 ppm after optimizing lung recruitment and hemodynamic support. The aim is to achieve an improvement in PaO_2_ by at least 20 mmHg; after that, slow weaning can be started (decrements by 5 ppm every 4 h until 5 ppm, then decrements by 1 ppm). Of note, doses > 20 ppm do not seem beneficial and iNO therapy should be interrupted to prevent side effects in infants failing to respond [41].

### 8.5. Steroids

Postnatal steroids reduce the inflammatory process, stabilize vascular membranes and enhance the cardiovascular stability in neonates [62,63,64]. Both nebulized (Budenoside) and systemic (Methylprednisolone) steroids have provided benefits in the duration of hospital stay, the duration of oxygen supplementation and radiological clearance in MAS patients [65]. Hydrocortisone improved oxygenation and systolic blood pressure in PPHN patients who were refractory to conventional treatment [66]. However, there is no conclusive evidence for the routine use of steroid therapy in infants with MAS; thus, further research is warranted [67,68,69].

### 8.6. Inotropic Therapy

MAS patients with hypotension or a reduced left cardiac output should receive inotropic therapy [41,70]. Of note, the caregiver should consider the effect of each cardiotonic agent on (i) systemic and pulmonary vascular resistance, (ii) ductal and atrial shunting and (iii) peripheral vasculature. Echocardiography is mandatory to assess the severity of the hemodynamic conditions and to choose the appropriate inotropic agent [46]:(a)If the echocardiographic features do not show a reduction of contractility and/or a reduction of the left ventricular output (LVO), hypotension is likely due to peripheral vasodilatation and vasopressors with action directed on systemic venous resistance (such as dopamine, norepinephrine or vasopressin);(b)If echocardiographic features show a low LV preload with RV/LV systolic dysfunction, positive inotropic agents with a pulmonary vasodilator effect (such as norepinephrine) are indicated; milrinone can be used in association with inotropes such as dobutamine or vasopressin because it causes pulmonary and systemic vasodilation;(c)If systemic blood pressure is stable, milrinone should be used in case of cardiac dysfunction; milrinone is a powerful vasodilator of the pulmonary circulation that also has a positive lusitropic and inotropic action while it also causes systemic vasodilation and reduces mean arterial pressure.

### 8.7. Extracorporeal Membrane Oxygenation (ECMO)

MAS patients failing conventional therapy (such as HFOV and iNO) require ECMO [71]. Current indications for ECMO are the occurrence of one the following conditions: (i) inadequate tissue oxygen delivery despite maximal therapy (rising lactate, worsening metabolic acidosis, the sign of end organ dysfunction); (ii) severe hypoxic respiratory failure with acute decompensation (PaO_2_ < 40 mmHg); (iii) an oxygenation index with sustained elevation and no improvement; iv) severe pulmonary hypertension with evidence of RV/LV dysfunction [72]. Although there is a decrease in the need for ECMO, a few MAS patients still require such treatment that results in a high survival (around 95%) [71,73,74,75].

### 8.8. Therapeutic Hypothermia

MAS patients should be offered therapeutic hypothermia when the criteria for asphyxia are met [76].

### 8.9. Therapeutic Considerations

The current treatment of MAS is exclusively supportive and current strategies do not act directly on the pathogenetic mechanism of lung damage. In fact, the development of specific therapies has been hindered by the poor understanding of MAS pathophysiology [4,77]. Further research may focus on molecules that could potentially interfere with the mechanisms underlying lung damage (such as apoptosis inhibitors and protease inhibitors).

## 9. Prognosis

MAS is associated with considerable morbidity and mortality. Among the pulmonary complications, air leak (i.e., pneumothorax or pneumomediastinum) occurs in 15–33% of MAS patients [78]. The mortality rate reduced from around 40% in the 1970s to below 5–12% in the last decade and is mainly associated with asphyxia and pulmonary hypertension [79]. Such a positive trend can be explained by the reduced incidence of post-maturity and the improvement in neonatal management at birth and in the NICU [10]. Despite the improved survival rate over the last decades, the long-term morbidity among survivors remains a major concern [80]. Survivors are at risk of pneumonia, a reduced functional residual capacity, bronchial hyperreactivity and asthma and about 5% of survivors still require oxygen therapy at one month of age [78]. MAS may also be associated with long-term neurodevelopment disabilities regardless of the mode of delivery and the treatment [81].

## 10. Prevention

Over the last few decades, the main prevention strategies for reducing the incidence of MAS have included: (i) the reduction of post-term births by inducing labor, (ii) the more aggressive management of deliveries when facing alterations in the cardiotocographic tracing and (iii) the improved management of the critical newborns in the delivery room [76,80,82]. Gastric lavage is not recommended for preventing MAS [11,83] while amnioinfusion may play a role in settings with limited peripartum surveillance [84].

## 11. Management of Infants Born through MSAF

The management of infants born through MSAF has greatly changed in the last few decades [11,85]. The first reference to tracheal suctioning dates back to 1960 when Dr. Stanley James postulated in a neonatal resuscitation textbook that “if meconium had been aspirated into trachea, it should be suctioned out” [86]. In 1974–1976, intrapartum oro-nasopharyngeal suctioning and endotracheal intubation and suctioning were routinely provided [54,55,87,88].

In 1987–1994, the initial evidence suggested that endotracheal intubation and suctioning should be offered only to non-vigorous infants [89] but international guidelines on neonatal resuscitation did not change [90,91,92].

In 2000, a large trial showed that intubation/tracheal suctioning was not beneficial in vigorous infants [49], resulting in the new recommendation of limiting endotracheal intubation and suctioning to non-vigorous infants [93,94,95].

In 2004, findings from another large trial showed that intrapartum oro-nasopharyngeal suctioning was not beneficial in full-term infants born through MSAF [35], leading to a change in the next edition of the guidelines on neonatal resuscitation (intrapartum oro-nasopharyngeal suctioning was no longer recommended; endotracheal intubation and suctioning was limited to non-vigorous infants) [95,96].

In 2010, endotracheal intubation and suctioning for non-vigorous infants was still recommended due to insufficient evidence to change practice [97] whereas the recommendation was overturned in 2015 (endotracheal intubation and suctioning for non-vigorous infants only if obstruction was suspected) due to insufficient evidence to continue this practice [31,98,99].

In 2020, this indication was confirmed in the last consensus on science and supported by a systematic review [76,100,101]. However, such an indication was based on a low certainty of evidence and the most appropriate approach is still a matter of debate [102]. Recent observational studies showed that the incidence or severity of MAS was not increased after the release of the last indication [22,103]. The main reasons leading to the last indication were potential procedure-related complications (such as apnea, bradycardia, airways and esophageal injuries, dislocation of the vocal cords, stridor) and the risk due to a delay in starting positive pressure ventilation. Nonetheless, a few studies have reported a very low incidence of such complications [31,49,89] and a recent manikin study showed a clinically irrelevant magnitude of the delay in starting positive pressure ventilation [104]. Of note, the presence of stained fluid during labor represents an alarm signal that requires careful monitoring of the cardiotocographic tracing and the presence of the neonatal team in the delivery room.

## 12. Conclusions

Despite progress in the knowledge of the pathogenesis, prevention and treatment, MAS remains a severe neonatal disease. Understanding the causes (inflammation, infection, hypoxia) triggering fetal bowel activity and disclosing the mechanisms contributing to the meconium passage in utero is warranted to improve MAS prevention. The current treatment of MAS patients is exclusively supportive and current strategies do not act directly on the pathogenetic mechanism of lung damage. Further, the specific role of the timing of the injury (antenatal, perinatal or postnatal) affecting the long-term neurodevelopmental and pulmonary outcome is still not well understood. An adoption of less invasive ventilation approaches to prevent pulmonary damage and the treatment of PPHN with newer agents (i.e., L-Citrulline, endothelin receptor antagonists) may have a role in preventing lung damage. The transfer of knowledge, approaches and equipment from high to low resource settings will play a crucial role in the global improvement of the management and outcome of MAS patients.

## Figures and Tables

**Figure 1 children-08-00230-f001:**
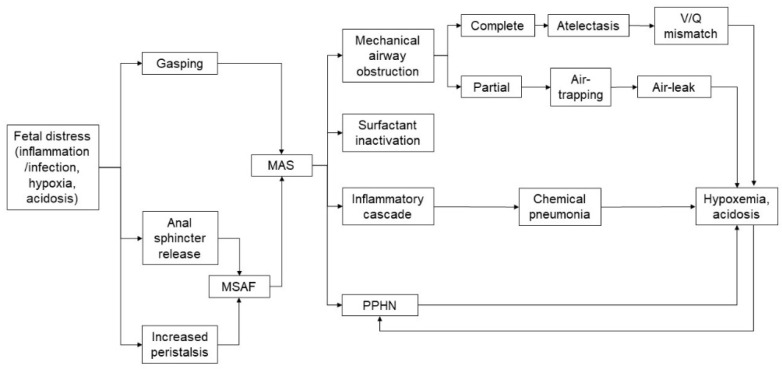
Pathophysiology of meconium aspiration syndrome (MAS). **LEGEND:** MAS, meconium aspiration syndrome; MSAF, meconium stained amniotic fluid; PPHN, persistent pulmonary hypertension of the neonate; V/Q mismatch, ventilation/perfusion mismatch.

**Figure 2 children-08-00230-f002:**
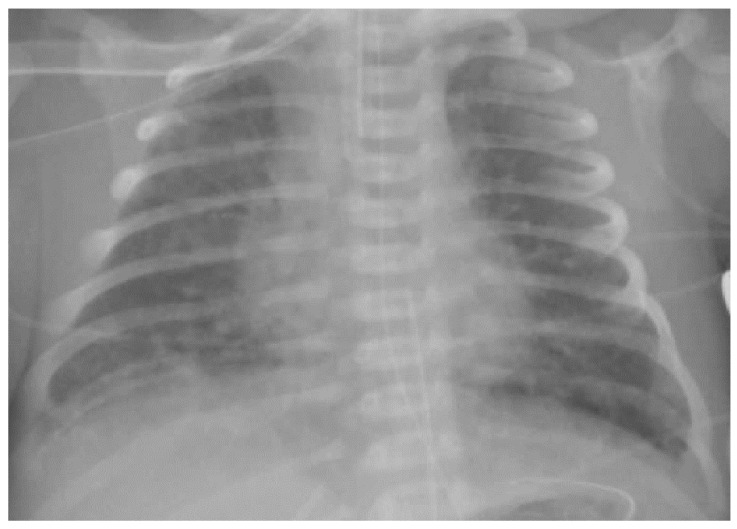
Chest X-ray of meconium aspiration syndrome.

## Data Availability

Not applicable.

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
