# Peer review of "Meconium Aspiration Syndrome: A Narrative Review"

_children, 2021, doi:10.3390/children8030230_

Round 1

Reviewer 1 Report

The author performed review of literature regarding meconium aspiration syndrome.

I cannot support publication of the review because of the following reasons:

  1. First of all this topic has been published many times and many similar reviews exist in medical literature (with the same title and concept). On the other hand there is also systematic review published on the same topic.
  2. A similar review with the same subtitles and the parts of the text has been published. So, what is new in this review?! REFERENCE: Sayad E, Silva-Carmona M. Meconium Aspiration. 2021 Jan 11. In: StatPearls [Internet]. Treasure Island (FL): StatPearls Publishing; 2021 Jan–. PMID: 32491357.
  3. The authors missed the point because this is more like a chapter from a book and not real review of literature. The authors should review crucial clinical factors/symptoms and perform review on each (compare findings from large series of patients)
  4. The conclusion that meconium aspiration syndrome is a rare clinical condition with a complex multifactorial pathophysiology which requires further investigation and that despite the improved survival rate over the last decades, long-term morbidity among survivors remains a major concern is well known and has been reported many times.
  5. Most important I do not see any benefit for the readers from this review because all presented information’s can be found in any basic pediatric book.
  6. This review can be written by one person. I do not see the role of the six authors at this paper. E.g. In authors contribution the author’s state that two authors were responsible for the methodology. What methodology? I do not see any methodology in this manuscript.

Reviewer 2 Report

Meconium aspiration syndrome is a real problem in neonates, and a comprehensive systemic review is always welcome. A review article should try to add the new and evolving science and try to give readers a concrete message. This review has a good beginning, but the author is failed to offer anything concrete for readers. I learned nothing new after reading this article, and also, there are a lot of inadequacies in the diagnostics and management part. here are suggestions:

1) Line 26-27: mentioning that MAS is uncommon in < 34 weeks is out of context here. Push it to the next paragraph; otherwise, it's a repetition.

2) pathophysiology:

  • In 20-30% of an infant with severe MAS: it happens before the infant is born. There are studies showing meconium in infants born stillbirth. I ask you to review that literature and add that to pathophysiology. Readers might be interested in knowing that. 
  • Infection: meconium itself might be sterile but could enhance the chances of E coli pneumonia. Please read and edit if needed. 

3) Clinical features: consider dividing between general findings and respiratory findings, and explain various respiratory findings to mild RD to traumatic ling damage to PPHN and ECMO need. 

Diagnosis: Provide more clarity. Clinical exam, chest x-ray, blood work for sepsis, and to assess respiratory severity like oxygenation index etc. then i would also mention Echo findings and its role in diagnosing PPHN. 

also look into the role of ultrasound.

Treatment: this aspect of a review article intrigues readers. This needs to be completely re-written: organized and give a definitive message to the readers at each management strategy's end, if possible. 

Again, you can divide treatment into general and respiratory. General can include things like temperature, blood pressure, cooling. At the same time, respiratory support should give guidance on when to intubate, switch to HFOV, administer surfactant or start iNO or call iNO refractory PPHN.

Steroids should be up after iNO management. Review use of steroid in MAS/PPHN. This is very important. You can't just write two lines, which are inconclusive.
Ionotropic support: in review, you want readers to read another article? I would give them a sense of your understanding. 
Readers would want to believe this MSAF; please review and concise. 

Round 2

Reviewer 1 Report

The authors revised manuscript. Unfortunately I do not see any significant improvement. As I pointed previously a similar review with the same subtitles and the parts of the text has been published. So, the question remains: what is new in this review? This is more like a chapter from a book and not real review of literature. The authors should review crucial clinical factors/symptoms and perform review on each (compare findings from large series of patients). All presented information’s can be found in any basic pediatric book. In my opinion this is not enough for publication in international journal with such impact factor.

Reviewer 2 Report

Authors have made necessary changes and this manuscript is good for the final review for grammar and spelling.

I would accept this article in its current form.
